# Admission Point-of-Care Testing for the Clinical Care of Children with Cerebral Malaria

**DOI:** 10.3390/tropicalmed9090210

**Published:** 2024-09-11

**Authors:** David Wichman, Geoffrey Guenther, Nthambose M. Simango, Mengxin Yu, Dylan Small, Olivia D. Findorff, Nathaniel O. Amoah, Rohini Dasan, Karl B. Seydel, Douglas G. Postels, Nicole F. O’Brien

**Affiliations:** 1School of Medicine and Health Sciences, The George Washington University, Washington, DC 20052, USA; 2Division of Infectious Diseases, Department of Pediatrics, Boston Children’s Hospital, Harvard Medical School, Boston, MA 02115, USA; 3Department of Paediatrics and Child Health, Kamuzu University of Health Sciences, Blantyre 3, Malawi; 4Department of Statistics and Data Science, The Wharton School, University of Pennsylvania, Philadelphia, PA 19104, USA; 5College of Arts and Sciences, University of Virginia, Charlottesville, VA 22903, USA; 6Blantyre Malaria Project, Kamuzu University of Health Sciences, Blantyre 3, Malawi; 7Department of Osteopathic Medical Specialties, College of Osteopathic Medicine, Michigan State University, East Lansing, MI 48824, USA; 8Division of Neurology, The George Washington University, Children’s National Hospital, Washington, DC 20010, USA; 9Division of Critical Care Medicine, Department of Pediatrics, Nationwide Children’s Hospital, College of Medicine, The Ohio State University, Columbus, OH 43205, USA

**Keywords:** malaria, pediatrics, point-of-care testing

## Abstract

Point-of-care testing (PoCT), an alternative to laboratory-based testing, may be useful in the clinical care of critically ill children in resource-limited settings. We evaluated the clinical utility of PoCT in the care of 193 Malawian children treated for World Health Organization-defined cerebral malaria (CM) between March 2019 and May 2023. We assessed the frequency of abnormal PoCT results and the clinical interventions performed in response to these abnormalities. We determined the association between abnormal PoCT results and patient outcomes. Overall, 52.1% of all PoCT results were abnormal. Of the children with abnormal results, clinical interventions occurred in 16.9%. Interventions most commonly followed abnormal results for PoCT glucose (100.0% of the patients had treatment for hypoglycemia), potassium (32.1%), lactate (22.0%), and creatinine (16.3%). Patients with hypoglycemia, hyperlactatemia, and hypocalcemia had a higher mortality risk than children with normal values. Future studies are needed to determine whether obtaining laboratory values using PoCT and the clinical response to these interventions modify outcomes in critically ill African children with CM.

## 1. Introduction

Compared with clinical settings in high-income countries, laboratory-based diagnostic tests are less commonly available in resource-limited settings (RLSs) [1]. Even if clinical laboratory-based testing is available, turnaround times may be long, limiting the ability of results to inform clinical care. Consequently, syndromic management, clinical decision-making informed by history and examination without support from laboratory testing, is often used in RLSs. Unfortunately, syndromic management may result in misdiagnosis, undertreatment, or overtreatment, potentially worsening outcomes [2].

Point-of-care testing (PoCT) is an alternative to laboratory-based testing that may be useful for acute patient management in RLSs. PoCT devices typically require minimal training, deliver results quickly, and may be comparatively inexpensive [3]. Real-time results from PoCT can guide timely clinical decision-making, including informing the need for additional laboratory-based diagnostic testing or changing a patient’s level of care within a hospital, thus conserving limited healthcare resources [4,5]. Increasingly, clinicians in RLSs rely upon PoCT to assess and manage critically ill patients. Thirty-two percent of clinicians in Nigeria reported using blood gas PoCT to inform clinical decision-making [6].

Malaria is a common cause of hospitalization and death worldwide. Cerebral malaria (CM), coma with malaria infection, is often accompanied by abnormalities in other organ systems [7]. The utility, timing, and duration of PoCT of blood glucose in African children with CM have recently been elucidated [8]. Similarly, others have demonstrated that blood lactate levels are a prognostic indicator of patient mortality [9,10,11]. Currently, little is known about the clinical utility of other PoCT tests in the recognition or management of pediatric patients with CM. It is unknown whether addressing abnormalities found with other PoCT affects clinical outcomes.

The Pediatric Research Ward (PRW) at Queen Elizabeth Central Hospital in Blantyre, Malawi, is a clinical research unit focused on improving the clinical care of children with CM. Participants in multiple observational and interventional studies undergo extensive diagnostic testing and are treated using standard protocols. These protocols include measurement for blood glucose and lactate via PoCT every 6 h for the first 24–48 h of hospitalization [12]. In 2019, clinicians began performing PoCT for creatinine and capillary blood gases in children admitted to the nearby Pediatric Intensive Care Unit (PICU). Creatinine and blood gas PoCT were expanded to all children admitted to the PRW beginning in February 2021.

Since the timing and duration of testing for both glucose and lactate have recently been evaluated, we aimed to assess the frequency of abnormal PoCT values of creatinine, lactate, glucose, and electrolytes in pediatric patients with CM. We aimed to evaluate how often these values resulted in changes in clinical management and the types of interventions that followed abnormal results. In addition, we aimed to evaluate associations between abnormal PoCT results and patient outcomes.

## 2. Materials and Methods

The three parent research studies of this project are prospective studies of children with World Health Organization (WHO)-defined CM admitted to the PRW and PICU at Queen Elizabeth Central Hospital. All children are 6 months to 12 years old and have clinical CM: *Plasmodium falciparum* parasitemia on peripheral blood smear, a Blantyre Coma Score (BCS) ≤2, and no other discernable cause of encephalopathy. At the time of enrollment in the parent study, the participants’ guardians provide written informed consent that includes possible future secondary analyses of the collected data. From these data, we performed a retrospective cohort study of those children enrolled in the parent studies between March 2019 and May 2023. The parent studies have varying treatment protocols, although all children received standard-of-care treatments for pediatric CM. The three parent studies were approved by the ethics committees of Michigan State University (USA), the University of Rochester (USA), and/or the University of Malawi College of Medicine (now Kamuzu University of Health Sciences).

Upon admission, children were clinically stabilized and immediately treated with intravenous artesunate according to Malawian Ministry of Health guidelines. During clinical stabilization, demographics and vital signs were collected, and children underwent a comprehensive physical examination. Finger-prick blood samples were collected to determine the malaria parasite species and density, packed cell volume (PCV) (roughly equivalent to hematocrit), as well as blood glucose and lactate concentrations (using Aviva Accu-Check (Zurich, Switzerland) and Arkray Lactate Pro 2 (Kyoto, Japan), respectively), all considered standard of care in our hospital unit. Additionally, we collected finger-prick samples for blood gas analysis (Abbott iSTAT, Chicago, IL, USA) and creatinine (Nova Biomedical StatSensor, Waltham, MA, USA). Immediately afterward, we collected venous blood for traditional laboratory-based electrolyte testing.

Patients received 20 mL/kg of whole blood if the admission PCV was less than 15% or if there were signs of circulatory compromise (defined as respiratory distress, a capillary refill time of >2 s, weak pulse, and/or cool extremities). Clinical or sub-clinical seizure activity, if identified by a 30-min routine EEG, was treated with 0.2 mg/kg of intravenous diazepam followed by intravenous phenobarbital 20 mg/kg if seizures continued. Additional antiseizure medications were administered at the discretion of the treating physician.

PoCT for creatinine and/or electrolytes was repeated after admission at the discretion of the treating clinician. Similarly, there were no unit-wide standardized interventions for abnormal laboratory results, physical exam findings, or vital signs.

At the time of hospital discharge, survivors underwent a neurological examination, and parents were asked if their child had returned to their pre-hospitalization neurological baseline. Children with normal examinations whose parents indicated their child had returned to their pre-hospital status were classified as alive and normal. All other survivors were classified as alive with neurologic sequelae. If the child died during hospital admission, they were classified as such.

A team of physicians experienced in the treatment of pediatric CM created a list of possible interventions that could logically have followed each abnormal PoCT result. This team also developed consensus reference ranges for each of the PoCT results (Appendix A). Patient charts were reviewed retrospectively by a team of clinical investigators who noted PoCT results and whether patients received the candidate interventions.

### Statistical Analysis

Variables were summarized using frequencies with percentages, medians with interquartile ranges, and means with standard deviations where appropriate. Differences between groups were assessed using Kruskal–Wallis tests for continuous and ordinal variables, and chi-square or Fisher exact tests for categorical variables. All analyses were conducted using R statistical software version 4 (R Foundation for Statistical Computing, Vienna, Austria). We did not control for multiple comparisons due to the exploratory nature of the analysis. Both PoCT and laboratory-based testing results were categorized as normal or abnormal on the basis of consensus reference ranges (Appendix A).

## 3. Results and Discussion

Between March 2019 and May 2023, 193 children with CM were admitted to the Pediatric Research Ward (Table 1) and enrolled in one or more of the three prospective parent studies. Parent study procedures included obtaining certain PoCT. Glucose and lactate measurements were taken for all 193 patients. PoCT creatinine levels were measured in 169 patients (88%), and electrolytes derived from blood gas were performed in slightly more than half of the patients. During the period of observation, 52.1% of all PoCT results were abnormal.

Bicarbonate was the most frequently measured value that was abnormal (76.2%), followed by lactate (61.1%) and creatinine (58.0%) (Table 2). Glucose (low 7.9%; high 38.0%) and sodium levels (30.6%) had the lowest percentage of abnormal results.

Abnormal PoCT results were followed by interventions in 16.9% of cases. Normal PoCT results were followed by one of the candidate interventions in 3.6% of cases.

The frequency of abnormal PoCT results likely reflects the systemic involvement that occurs in severe malaria infection [7,13]. Although the frequency of candidate interventions was low after results of several types of PoCT testing became known, children with abnormal results were more likely than those with normal results to receive interventions.

Patients who died were more likely than those who survived to have received interventions for abnormal PoCT results. This association likely represents more critical illness on presentation leading to more interventions for those who ultimately died, rather than an adverse effect of intervention. We believe that children with more critical illnesses may have been more likely to receive multiple interventions regardless of the PoCT results. Nevertheless, it is also clear from our analysis that patients with abnormal PoCT results were more likely to receive an intervention directed at such results than those with normal results. This suggests that there is likely clinically actionable information gleaned from PoCT. Further studies are needed to determine whether there is a causal association between interventions for abnormal PoCT results and a change in mortality or morbidity risk. The results of such studies may allow the development of richer diagnostic and treatment algorithms for children with CM or other critical illnesses.

### 3.1. Bicarbonate

Of all PoCT results, bicarbonate was the most frequently abnormal (76.2%). Abnormal bicarbonate was followed by sodium bicarbonate bolus administration in 5% of children with abnormalities (Table 3).

PoCT devices for bicarbonate, such as blood gas assays, are among the most expensive and least available PoCT devices in many RLSs [6], although less expensive versions are being developed for use in RLSs [3]. Low bicarbonate is associated with poor prognosis in patients with CM [14]. The role of bicarbonate therapy to aid in the reversal of malaria-associated lactic acidosis is controversial [15].

Despite the high frequency of abnormal results (>75%), the rate of intervention following abnormal bicarbonate levels was low (5.0%). Intervention with intravenous bicarbonate may have been limited in this study due to an intermittent nonavailability of intravenous bicarbonate at our hospital.

### 3.2. Lactate

Abnormal lactate measurements were common (61.1%). The most frequent clinical intervention performed after receiving an abnormal PoCT lactate result was the administration of a blood transfusion if the child was anemic. Several children who were not anemic and had cold extremities, a capillary refill time greater than 2 s, or hypotension received a normal saline fluid bolus and/or initiation of an intravenous epinephrine infusion. One or more of these interventions followed abnormal lactate test results in 22.0% of children with abnormal lactate results. Levels of lactate were higher in those who received interventions (median 9.99 mmol/L; IQR 5.50–13.60) compared with those who, despite having abnormal levels, did not receive interventions (median 7.15; IQR 4.50–8.90) (*p* = 0.008).

The pathogenesis of elevated lactate in patients with CM is likely multifactorial: a combination of parasite metabolism with a lactate byproduct as well as anaerobic metabolism in host tissues due to sequestration of parasitized red blood cells and subsequent microvascular occlusion. Metabolic stress may be further exacerbated by increased metabolic demand on the liver, reducing lactate clearance [16,17,18].

Elevated lactate on PoCT is associated with a poor prognosis in several populations in sub-Saharan Africa [12,19,20,21], including children with CM [9]. The optimal intervention strategy (if any) to be implemented in response to high PoCT lactate values in children with CM remains unknown. As per the parent study protocol, patients received a blood transfusion regardless of their lactate level if their admission PCV was less than 15% or if they had hemodynamic instability, which is why four patients with normal PoCT lactate levels received blood transfusions.

Given the association between hyperlactatemia and increased mortality risk, it is unsurprising that lactate was one of the most common PoCT results followed by clinical intervention. Further research is needed to clarify if there is an optimal lactate threshold that should prompt a clinical intervention and what that intervention should be.

### 3.3. Creatinine

Many patients in our cohort (58.0%) had abnormal creatinine PoCT values. After receipt of creatinine results, 16.3% of these children received a candidate intervention. The most frequent interventions were a change in the rate of fluid administration, the minimization of nephrotoxic medications, or the administration of furosemide. The test results did not differ significantly between those who did (median 0.95 mg/dL; IQR 0.76–1.31) and did not (median 0.95 mg/dL; IQR 0.77–1.08) receive interventions (*p* = 0.486).

Acute kidney injury (AKI) is common in children with severe malaria, occurring in 24–59% of cases due to reduced renal perfusion, inflammation, and parasite or hemolysis-mediated damage [22]. The presence and degree of AKI is associated with adverse outcomes [23,24,25,26] in children with severe malaria. Whether and how elevated creatinine PoCT results affect clinical care in children with severe malaria remains unknown.

PoCT creatinine was shown to underestimate rates of AKI in one study of young children with severe malaria compared with laboratory-based testing [27]. The high rate of abnormal creatinine values (58%) among our patients is congruent with AKI rates previously reported in similar patient groups [22,26,27]. The relatively low rate of subsequent interventions (16%) following abnormal creatinine is likely explained by the fact that mild AKI (median creatinine 0.95 mg/dL; IQR 0.76–1.31) was commonly encountered in our study cohort. This likely resulted in conservative measures, such as the clinical observation of urine output, without other interventions.

### 3.4. Glucose

Using the WHO-defined threshold for the treatment of hypoglycemia associated with severe malaria of 3.0 mmol/L [28], PoCT glucose values on enrollment were abnormally low in 7.3% of patients (n = 14). Of these, 100% received a bolus of dextrose-containing fluids. Using the consensus-determined upper limit of normal of 6.0 mmol/L, 38.0% (n = 73) of patients had abnormally high glucose levels. Of these, only three patients (4.1%) had a glucose-related intervention (a decrease in the dextrose concentration of maintenance fluids). One patient with a normal admission glucose level later developed hypoglycemia requiring a bolus of dextrose-containing fluids.

Hypoglycemia has long been known to be associated with poor prognosis in children with CM [11]. A recent systematic review of prognostic models in pediatric malaria showed that half of such models included hypoglycemia, demonstrating that it is one of the strongest prognostic indicators available [29]. The prognostic importance and optimal duration of blood glucose PoCT measures in CM have recently been reported [8].

Our results are congruent with previously published literature. PoCT glucose values on enrollment were abnormal in many patients, and hypoglycemia was strongly associated with increased mortality risk. In those with glucose values below the standard treatment threshold for intervention (glucose < 3.0 mg/dL), all patients received a bolus of dextrose-containing fluids, as recommended by the WHO. Glucose values were abnormally high in 38.0% of children, but these did not commonly result in an intervention. It is possible that this represents a lack of a consensus approach to treatment thresholds for hyperglycemia in critically ill children.

### 3.5. Electrolytes

Abnormal PoCT electrolyte levels of potassium, ionized calcium, and sodium were noted in 49.1%, 39.6%, and 30.6% of our patients, respectively. Of these, the most common PoCT result that preceded an intervention was potassium, for which 32.1% of patients with abnormal values received an intervention. The most common interventions after abnormal potassium results were changing maintenance fluids from lactated ringers to normal saline or administering furosemide. Nine percent of patients with abnormal sodium levels received enteral NaCl to correct hyponatremia. Ionized calcium levels were followed by intervention in 7.5% of patients with abnormal PoCT levels, most commonly the administration of a calcium bolus.

Hypercalcemia and hyperkalemia, as determined by laboratory-based testing, were the most commonly observed electrolyte derangements in one study of 56 Kenyan children with severe malaria, one-third of whom had CM [30]. Another study of 38 Kenyan children showed that most had normal potassium on admission, but many developed hypokalemia shortly after hospitalization, possibly due to intravenous volume resuscitation [31]. Several studies have demonstrated that mild hyponatremia is common in severe malaria [32,33,34]. Hypocalcemia is common in malaria infections across the spectrum of severity [35,36,37]. Little is known about the clinical or prognostic importance of these laboratory abnormalities.

It is not surprising that, given the well-established relationship between elevated potassium levels and cardiac function, interventions related to potassium were the most common in this study among patients with electrolyte abnormalities.

### 3.6. Comparison of PoCT to Laboratory-Based Testing

We compared the results from PoCT testing to traditional laboratory-based testing (Appendix A). The results were similar between the two methods for potassium (*p* = 0.708). The results differed for bicarbonate (*p* < 0.001) and sodium (*p* < 0.001), although the medians and middle interquartile range results showed similar attributes when compared to a normal range (i.e., below normal for bicarbonate and within the normal range for sodium). Measurements of calcium could not be directly compared, as PoCT measured ionized calcium, while traditional laboratory-based testing assessed total calcium. As such, statistical comparison between the testing modalities was based on the proportion of results within their respective normal ranges. These proportions did not significantly differ between testing modalities (PoCT vs. laboratory-based testing).

### 3.7. Outcomes

Of the children included in this study, 29 (15.0%) died, 135 (69.9%) were alive without neurologic sequelae, and 29 (15.0%) were alive with neurologic sequelae (Table 4). On univariate analysis, admission hypoglycemia, hyperlactatemia, and hypocalcemia, as determined by PoCT, were each associated with increased mortality (*p* = <0.001, *p* = 0.017, *p* = 0.013, respectively) (Table 5).

Laboratory-based testing revealed similar relationships between results and outcomes. Patients with low bicarbonate, hyperkalemia, and hypocalcemia were at increased mortality risk. Although hyponatremia was not associated with mortality in either laboratory-based testing or PoCT, it was associated with a lower frequency of neurological morbidity in survivors using both laboratory and PoCT testing results.

Our results add to the well-established body of evidence which suggests that hyperlactatemia and hypoglycemia are each associated with higher mortality in patients with malaria [10,11,29,33,38]. In our study cohort, there was an increased proportion of deaths in children admitted with hyperglycemia on PoCT (15.1%) than in children who had normal glucose results (9.5%), but this difference did not reach statistical significance (*p* = 0.373).

Hypocalcemia and hyperkalemia were both associated with a higher risk of death in our cohort. The limited number (n = 3) of patients who received an intervention after hypocalcemia was detected by PoCT had exceedingly low calcium levels (median = 0.85) and poor outcomes (either death or survival with sequelae). Although this study suggests an association between poor outcomes and abnormal PoCT calcium levels, the clinical impact of correcting this electrolyte is not well understood. The prospective assessment of calcium repletion and its impact on clinical outcomes is warranted.

Lower sodium levels on either PoCT or laboratory-based testing are associated with a lower likelihood of neurologic sequelae in survivors. The etiologies for hyponatremia in this population remain unclear, but given this association, one possible explanation is that children with hyponatremia were more likely to have undergone fluid resuscitation prior to admission since most resuscitation in our region is conducted with Ringer’s Lactate (Na concentration 134 meq/L). Since we could not objectively determine what interventions children may have received prior to admission to our hospital unit (in District Hospitals where PoCT is unavailable), these pre-unit interventions were not included in our analyses. Future studies should consider enrolling participants from the onset of their clinical illness.

We did not find a meaningful relationship between PoCT creatinine levels and mortality or neurologic morbidity. Our results are not congruent with those of other published observational studies of children with severe malaria, which have shown that the degree of AKI is associated with adverse outcomes [23,24,26]. It is possible that our candidate interventions played a role in decreasing adverse outcomes among those with AKI, although only a small percentage of abnormal values (16.3%) were followed by a candidate intervention. Alternatively, the degree of AKI was relatively mild in our cohort of children, so it may not have negatively impacted outcomes.

### 3.8. Strengths, Limitations, and Future Directions

This study has many strengths. To our knowledge, ours is the largest study of PoCT in pediatric CM and one of the largest studies of a broad array of PoCT in any critically ill pediatric population in Africa. Additionally, the clinical care pathways and personnel caring for participants did not change over time, decreasing the likelihood of temporal bias. Our study has several limitations. During review, our assessment of the relationship between clinical decision-making and abnormal PoCT results was limited by the retrospective nature of the study. Interventions we did not consider as logical to follow the PoCT result in question may have occurred and not been recorded in our analyses because the linkage between the abnormal PoCT result and the intervention may not have been stated in the medical record. Conversely, we may have overestimated the number of interventions that occurred due to PoCT, as clinicians were not always explicit about their reasons for performing interventions in their documentation. Our analysis may have not captured pre-admission interventions, which may have impacted outcomes.

Future prospective studies of PoCT in the clinical care of children with CM should consider interviewing clinicians directly about their decision-making after the receipt of PoCT results. Future clinical trials could also compare outcomes of children with CM undergoing routine PoCT versus standard of care. Certainly, if it is eventually found that the implementation of PoCT leads to better outcomes in children with CM, a future cost–benefit analysis may be warranted to evaluate the wide deployment of PoCT in the treatment of pediatric patients with CM in RLSs.

## 4. Conclusions

Clear guidelines exist for the management of hypoglycemia in pediatric CM. Currently, other abnormal PoCT results are not often followed by interventions in our hospital unit, despite the high frequency with which they occur and their association with poor outcomes. Further study is needed to determine whether children with CM would benefit from clinical interventions targeting these abnormal results.

## Figures and Tables

**Table 1 tropicalmed-09-00210-t001:** Demographics, history, hospital course, point of care testing, and clinical investigations (n = 193).

	Study Patients (n = 193)
Demographics	
Age (months), mean (SD)	57 (30.6)
Male, n (%)	104 (53.9%)
Vital signs/physical examination	
Temperature (°C), median (IQR)	38.6 (37.9, 39.4)
Heart rate (beats/min), median (IQR)	142 (128, 158)
Respiratory rate (breaths/min), median (IQR)	38 (28, 43)
Oxygen saturation (%), median (IQR)	95 (94, 98)
Systolic blood pressure (mmHg), median (IQR)	106 (97, 115)
Blantyre coma score, n (%)	
0	22 (11.5%)
1	64 (33.3%)
2	106 (55.2%)
MUAC (cm), median (IQR)	15.4 (14.5, 16.5)
Capillary refill time > 2 s, n (%)	15 (7.8%)
Respiratory distress (composite) *, n (%)	50 (25.9%)
Hospital course	
Duration of hospital stay (days), median (IQR)	5.9 (3.0, 6.0)
Other laboratory-based testing	
Admission platelet count (/mcL), median (IQR)	102,233 (38,000, 129,000)
Admission packed cell volume (%), median (IQR)	25.7 (21.0, 30.0)
Admission log_10_ parasite density (parasites/mcL), median (IQR)	3.096 (2.380, 5.137)
HIV reactive, n (%)	4 (2.4%)
Positive blood culture, n (%)	12 (6.5%)

n = number; SD = standard deviations; IQR = interquartile range; mmHg = millimeters mercury; * respiratory distress indicated if the clinician noted the presence of chest indrawing, intercostal retraction, deep breathing, nasal flaring, or grunting on admission.

**Table 2 tropicalmed-09-00210-t002:** PoCT results and patient outcomes.

PoCT	Normal		If Normal, Intervention Received		Abnormal		If Abnormal, Intervention Received	
Glucose	105	54.7%	1	0.0%	-	-	-	-
Low (<3.0 mmol/L)	-	-	-	-	14	7.3%	14	100%
High (>6.0 mmol/L)	-	-	-	-	73	38.0%	3	0.0%
Lactate	75	38.9%	7	9.3%	118	61.1%	26	22.0%
Creatinine	71	42.0%	5	7.0%	98	58.0%	16	16.3%
Bicarbonate	25	23.8%	0	0.0%	80	76.2%	4	5.0%
Sodium	75	69.4%	1	1.3%	33	30.6%	3	9.1%
Potassium	55	50.9%	3	5.5%	53	49.1%	17	32.1%
Ionized calcium	61	60.4%	0	0.0%	40	39.6%	3	7.5%
	467	47.9%	17	3.6%	509	52.1%	86	16.9%

**Table 3 tropicalmed-09-00210-t003:** Frequency of interventions.

Intervention	All	Abnormal PoCT Only
Blood transfusion given	22	18
Dextrose bolus given	16	15
Furosemide given	15	13
Fluid bolus given	12	8
Repeated PoCT monitoring *	11	8
Electrolyte modification: lactated ringers to normal saline	7	7
Epinephrine drip given	7	6
Changed glucose concentration in maintenance fluids	4	4
Low potassium diet	4	4
Calcium bolus given	3	3
Enteral sodium replacements given	3	3
Oral sodium replacements given	2	0
Bicarbonate treatment administered	2	1
Minimize nephrotoxic medications	1	1
Increased fluid rate	1	1
Free water deficit calculated and replacements started	1	0

* Patients were counted more than once if multiple PoCT was repeated.

**Table 4 tropicalmed-09-00210-t004:** Intervention frequency and outcomes for abnormal PoCT results.

PoCT	If PoCT Abnormal											
	Alive				Alive + Sequelae				Dead			
	Total		Received Intervention		Total		Received Intervention		Total		Received Intervention	
Glucose: low (<3.0 mmol/L)	5	35.7%	5	100%	1	7.1%	1	100%	8	57.1%	8	100%
Glucose: high (>6.0 mmol/L)	51	67.0%	1	0.0%	11	15.1%	1	9.1%	11	15.1%	1	9.1%
Lactate	75	63.6%	15	57.7%	18	15.3%	4	15.4%	25	21.2%	7	26.9%
Creatinine	62	63.3%	6	37.5%	18	18.4%	6	37.5%	18	18.4%	4	25.0%
Bicarbonate	48	60.0%	1	25.0%	15	18.8%	0	0.0%	17	21.3%	3	75.0%
Sodium	20	60.6%	3	100.0%	9	27.3%	0	0.0%	4	12.1%	0	0.0%
Potassium	29	54.7%	11	64.7%	13	24.5%	5	29.4%	11	20.8%	1	5.9%
Ionized calcium	21	52.5%	0	0.0%	8	20.0%	2	66.7%	11	27.5%	1	33.3%
	311	61.1%	42	13.5%	93	18.3%	19	20.4%	105	20.6%	25	23.8%

**Table 5 tropicalmed-09-00210-t005:** Patient outcomes by PoCT and laboratory-based testing results.

	Alive (n = 135)	Alive with Sequelae (n = 29)	Died (n = 29)	*p*-Value for Difference (Alive vs. Dead)	*p*-Value for Difference (in Survivors, Normal vs. Abnormal)
Point of care testing results					
Glucose (mmol/L)	5.700 (4.600, 6.700)	5.300 (4.900, 6.500)	4.80 (2.80, 6.70)	0.074	0.853
Lactate (mmol/L)	3.80 (2.40, 6.50)	3.70 (2.60, 7.20)	6.50 (4.10, 10.40)	0.001	0.681
Creatinine (mg/dL)	0.680 (0.540, 0.980)	0.710 (0.550, 1.040)	0.820 (0.705, 0.950)	0.239	0.563
Bicarbonate (mmol/L)	20.50 (17.65, 23.45)	18.40 (16.85, 22.02)	14.70 (12.10, 17.15)	<0.001	0.348
Sodium (mmol/L)	140.0 (137.0, 143.0)	142.00 (139.8, 147.5)	137.00 (0.00, 143.00)	0.798	0.020
Potassium (mmol/L)	4.500 (4.125, 4.800)	4.800 (3.975, 5.025)	4.700 (4.300, 5.025)	0.569	0.234
Ionized calcium (mmol/L)	1.170 (1.110, 1.240)	1.150 (1.087, 1.202)	1.080 (0.935, 1.165)	0.013	0.260
Laboratory-based testing results					
Bicarbonate (mmol/L)	16.0 (13.0, 18.0)	15.00 (11.50, 19.50)	10.50 (8.00, 14.25)	<0.001	0.489
Sodium (mmol/L)	137.0 (134, 140.0)	139.9 (138.0, 142.5)	139 (134.5, 141.0)	0.536	0.007
Potassium (mmol/L)	4.400 (3.950, 5.100)	4.500 (4.100, 4.850)	5.300 (4.550, 6.000)	0.010	0.909
Total calcium (mmol/L)	2.110 (2.010, 2.190)	2.100 (1.995, 2.215)	1.990 (1.755, 2.185)	0.032	0.715

## Data Availability

The datasets generated during the current study are available from the corresponding author upon reasonable request. Clinical and demographic data derived from the parent study are not the property of the authors and cannot be released by them.

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
