# Peer review of "Admission Point-of-Care Testing for the Clinical Care of Children with Cerebral Malaria"

_tropicalmed, 2024, doi:10.3390/tropicalmed9090210_

Round 1

Reviewer 1 Report

Comments and Suggestions for Authors

Well-designed and written manuscript with a clear analysis of PoCT data in relation of outcomes and treatments. Very informative for others in this field of research.

Specific points:

PCV was determined on the patients upon admission. These data were not included in the analysis although it must have informed interventions such as blood transfusion and have an important impact on the outcomes.

Minor issue: It is unclear why in Table 3 some children with normal PoCT received the intervention. For example, 22 patients received blood transfusion, but only 18 had abnormal PoCT. What was the decision to provide blood transfusion to those 4 patients based on? Was it based on PCV?

Author Response

Comments 1: PCV was determined on the patients upon admission. These data were not included in the analysis although it must have informed interventions such as blood transfusion and have an important impact on the outcomes.

Response 1: Lines 203 – 206. This is addressed in the following point.

Comments 2: Minor issue: It is unclear why in Table 3 some children with normal PoCT received the intervention. For example, 22 patients received blood transfusion, but only 18 had abnormal PoCT. What was the decision to provide blood transfusion to those 4 patients based on? Was it based on PCV?

Response: Lines 203 – 206. Thank you for pointing this out. We have clarified this difference in the updated manuscript, as these 4 children were hemodynamically unstable and thus received a transfusion regardless of PCV and PoCT results.

Reviewer 2 Report

Comments and Suggestions for Authors

As atttached

Comments on the Quality of English Language

Moderate editing required

Author Response

Comments 1: The title: “Admission Point of Care Testing for the Clinical Care of Children with Cerebral Malaria “ does not seem to reflect the aim of the study. For instance, “Impact of Errors in Point of Care Testing on the Clinical Outcomes in Children with Cerebral Malaria”. 
Response 1: Thank you for this suggestion. We believe our current title succinctly summarizes our aims, which were to evaluate the impact of normal and abnormal PoCT results on clinical care as well as associations between PoCT results and patient outcomes. 

Comments 2: Lines 54-5: Reference this “Compared to clinical settings in high-income countries, laboratory-based diagnostic tests are less commonly available in resource-limited settings (RLS).” 
Response 2: Line 50. We have included a reference to support our statement. 

Comments 3: Lines 54-60: No references at all. Please address this 
Response 3: Line 55. We have included a reference to support our statement.

Comments 4: Lines 87-92: The aim(s) of the study should be clearly stated here. Currently, it is a bit clumsy. In line 88, instead of “we assessed the frequency”, use “we aimed to assess the frequency”. In line 89, “We evaluated” should be changed to “We aimed to evaluate”. Apply same in line 91 
Response 4: Lines 83, 84, and 86. Thank you for these suggestions; we have included these suggestions in our latest manuscript.

Comments 5: Since the study was a retrospective one, how did the authors get the consent of the guardians of the children enrolled in the study? 
Response 5: Lines 89 – 101. Thank you for pointing this area of potential confusion out. We have reworded the first paragraph of the “Materials and Methods” section in our manuscript to better clarify this point. We conducted a retrospective cohort study based on data collected from three parent research studies which were prospective in nature. At parent study enrollment, the parent / guardian of participants provided informed, written consent which included the possibility of future secondary analysis of any data collected. 

Comments 6: The research design- retrospective cohort study- does not align with methods described here. This research design involves collecting data from existing records and the data will be analyzed. From the method described here, the authors described prospective studies where they collect the data directly from the patients/participants. It is indeed contradictory to the proposed research design- a retrospective cohort study. 
Response 6: Lines 89 – 101. As mentioned in the previous point, we have reworded the first paragraph of the “Materials and Methods” section to clarify the design of our study and data collection.

Comments 7: Were these results obtained prospectively or retrospectively? 
Response 7: Lines 144 – 146. We have reworded this section to clarify that we retrospectively analyzed data collected from three prospective parent studies.

Comments 8: Line 149: The table should be place just below where it was first mentioned for easy access by the reader. Apply the same to all the tables in the work. 
Response 8: We are not opposed to this change and will request that the editors update the placement of the tables, if possible.

Comments 9: Line 200: Table S2 should be removed from supplementary data and placed just below line 208. 
Response 9: Table S2 contains data comparing point-of-care testing (PoCT) results with laboratory-based testing results. Although we wanted to acknowledge differences in these testing modalities, we feel that placing this table in our manuscript would be a distraction from the primary analysis and outcomes that are novel to this manuscript. As such, we would prefer for this table to remain available as supplementary material.

Comments 10: Line 230-1: This sound like speculation, it is not acceptable in discussion. In this section, the authors should compare and contrast this work previous works 
Response 10: Lines 163 and 164. We have updated our wording to reflect our understanding that children with more critical illness may have been more likely to receive multiple interventions.

Comments 11: Line 239-243: Not properly discussed. No references. The authors should look at the state of the facilities for measurement of bicarbonate in resource-limited centres and its impact on POC testing. 
Response 11: Lines 176 – 180. We have revised our manuscript to include commentary with references related to PoCT bicarbonate measurement in resource-limited settings. 

Comments 12: Line 325: Your research design is not observational, it is retrospective. So there may not be enough basis for comparison. 
Response 12: Lines 321 – 323. We agree with this statement, and we have revised the manuscript to appropriately reflect the comparison of our results with other studies.

Comments 13: No conclusion to state the takeaway from the study. 
Response 13: Lines 350 – 354. We have added a “Conclusions” section to summarize the key findings and takeaways from our study.

Comments 14: I would recommend a merger of the results and discussion sections for easier flow in the work. The research design- retrospective cohort study does not tally with the design presented here.
Response 14: Thank you for this suggestion. We have combined the results and discussion sections in the latest revision.

Round 2

Reviewer 2 Report

Comments and Suggestions for Authors

The title
“Admission Point of Care Testing for the Clinical Care of Children with Cerebral Malaria “ does not seem to reflect the aim of the study. For instance, “Impact of Errors in Point of Care Testing on the Clinical Outcomes in Children with Cerebral Malaria”. Not addressed

Comments on the Quality of English Language

Moderate editing required